# Analysis of Catapult-Assisted Takeoff of Carrier-Based Aircraft Based on Finite Element Method and Multibody Dynamics Coupling Method

**Haoyuan Shao [1], Daochun Li [1,2], Zi Kan [1,*], Shiwei Zhao [1], Jinwu Xiang [1,2] and Chunsheng Wang [3]**

1   School of Aeronautic Science and Engineering, Beihang University, Beijing 100191, China; hyshao_14@buaa.edu.cn (H.S.); lidc@buaa.edu.cn (D.L.); shiweizhao@buaa.edu.cn (S.Z.); xiangjw@buaa.edu.cn (J.X.)
2   Tianmushan Laboratory, Hangzhou 310023, China
3   China Aero Poly-Technical Establishment, Beijing 100191, China; wcsxgd@163.com
*   Correspondence: kanzi2017@buaa.edu.cn

**Abstract:** Catapult-assisted takeoff is the initiation of flight missions for carrier-based aircrafts. Ensuring the safety of aircrafts during catapult-assisted takeoff requires a thorough analysis of their motion characteristics. In this paper, a rigid–flexible coupling model using the Finite Element Method and Multibody Dynamics (FEM-MBD) approach is developed to simulate the aircraft catapult process. This model encompasses the aircraft frame, landing gear, carrier deck, and catapult launch system. Firstly, reasonable assumptions were made for the dynamic modeling of catapult-assisted takeoff. An enhanced plasticity algorithm that includes transverse shear effects was employed to simulate the tensioning and release processes of the holdback system. Additionally, the forces applied by the launch bar and holdback bar, nonlinear aerodynamics loads, shock absorbers, and tires were introduced. Finally, a comparative analysis was conducted to assess the influence of different launch bar angles and holdback bar fracture stain on the aircraft's attitude and landing gear dynamics during the catapult process. The proposed rigid–flexible coupling dynamics model enables an effective analysis of the dynamic behavior throughout the entire catapult process, including both the holdback bar tensioning and release, takeoff taxing, and extension of the nose landing gear phases. The results show that higher launch bar angle increase the load and extension of the nose landing gear and cause pronounced fluctuations in the aircraft's pitch attitude. Additionally, the holdback bar fracture strain has a significant impact on the pitch angle during the first second of the aircraft catapult process, with greater holdback bar fracture strain resulting in larger pitch angle variations.

**Keywords:** catapult launch; carrier-based aircraft; holdback release; rigid–flexible coupling model; dynamic analysis

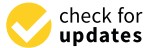

## 1. Introduction

Aircrafts taking off from aircraft carriers can employ various methods, including vertical takeoff, ski-jump takeoff, and catapult-assisted takeoff [1,2]. Among these, catapult-assisted takeoff is expected to be the primary choice for future carrier-based aircrafts. In this method, the aircraft accelerates rapidly, reaching speeds of around 270 km per hour within a distance of less than 100 m, all achieved in under 3 s [3]. In the 1960s, the United States conducted extensive experimental research on carrier-based aircraft catapult processes [4]. The Langley Research Center [5] conducted experiments on the rolling friction coefficients of aircraft landing gear tires on concrete runways and non-skid carrier decks. Berman [6] conducted simulated catapult fatigue tests on critical structural components of the E-1/C-1 aircraft to enhance their structural integrity. Michael [7] analyzed factors influencing carrier-based aircrafts' minimum takeoff speed, both with and without engine augmentation, based on data from flight tests of the F/A-18E/F aircraft. They also provided an analysis of the

acceleration on the launch end speed. These experimental and research efforts aimed to enhance the understanding and safety of catapult operations for carrier-based aircrafts.

Catapult-assisted takeoff is a multifaceted dynamic process with various interrelated systems and degrees of freedom. Zhu established a dynamic model for the launch bar and examined how parameters like mass and center of gravity affect its dynamic performance during aircraft catapult operations [8]. Additionally, a mathematical model of the steam catapult system was developed, and optimizations were made to parameters related to the wet steam accumulator [9]. The aircraft's nose landing gear is linked to the catapult shuttle through the launch bar, forming a coupled dynamic system that responds to the tensioning and release of the holdback bar. Wilson [10] conducted ship suitability tests for the F-35C, which included land-based trials and discussions on the outcomes of aircraft catapult and arrestment tests. Wang [11] presented a modeling technique based on a multi-agent system for carrier-based aircraft catapult processes, and the models of the landing gear and the catapult system have been simplified. Zhen [12] built a nonlinear steam catapult-assisted takeoff model of a carrier-based aircraft which considered the influences of the preset control surfaces, flight deck motion, ship bow airflow, and control system.

To accurately simulate the aircraft's holdback and taxiing process on a carrier deck, it is essential to consider the structural integrity of the holdback bar and the dynamic characteristics of the landing gear. Nie [13] established a six-degree-of-freedom dynamic model for the catapult process of carrier-based aircrafts, accounting for off-center aircraft positions. Qu [14] built an integrated system simulation model that incorporates the complex interactions among the carrier, aircraft, landing gears, as well as factors like wind fields from the aircraft carrier, deck command decisions, and pilot control policies. With the widespread application of computational multibody dynamics and virtual prototyping techniques, researchers have shifted their focus from fundamental dynamics equations to numerical computational methods. Current research on catapult-assisted takeoff dynamics primarily concentrates on specific phases of the process and the modeling of individual components within the catapult system. There is limited research on the comprehensive dynamics of the entire aircraft catapult process and the coupled dynamics of carrier-based aircrafts and catapult systems. Chen [15] established a catapult dynamics model of a carrier-based aircraft based on the multibody method, and a variable topology solution was carried out by adjusting dynamic augmentation equations. Dong [16] established a multi-body model for aircraft steam catapult systems using natural coordinate methods and topological analyses of the multi-body catapult launch system.

Numerical simulations provide a more detailed and comprehensive approach to assessing dynamic responses compared to mathematical models based on general mechanics equations. Therefore, numerical simulation methods are suited for load assessments to fulfill the demands of modern, refined carrier-based aircraft and catapult system analyses. In this study, a rigid–flexible coupling dynamic modeling method is presented for the catapult-assisted takeoff process. The model comprises the electromagnetic catapult mechanics, tensioning and release mechanism, and aircraft landing gear dynamics. The investigation focuses on the effects of varying holdback bar fracture strains and launch bar angles on the aircraft's attitude during the catapult process, as well as the dynamic response of the landing gear.

## 2. Modeling Approach

### 2.1. Dynamic Equations

The structural components are described by primarily 8-node hexahedral elements and 4-node shell elements to describe the structural components. The dynamic equation of the structural component is

$$M\ddot{u}^{t+\Delta t} + C\dot{u}^{t+\Delta t} + Ku^{t+\Delta t} = F^{t+\Delta t} \tag{1}$$

where $M, C, K$ are the mass, damping, and stiffness matrices, respectively, $F^{t+\Delta t}$ is the external load at time $t + \Delta t$. $\ddot{u}^{t+\Delta t}, \dot{u}^{t+\Delta t}, u^{t+\Delta t}$ are the displacement, velocity, and acceleration matrices, respectively.

The dynamic equation for each finite element is

$$m_i \ddot{u}_i^{t+\Delta t} + c_i \dot{u}_i^{t+\Delta t} + k_i u_i^{t+\Delta t} = f_i^{t+\Delta t} \tag{2}$$

where $m_i$, $c_i$, $k_i$ are the mass, damping, and stiffness matrices, respectively, and $f_i^{t+\Delta t}$ is the external load at time $t + \Delta t$. $\ddot{u}_i^{t+\Delta t}, \dot{u}_i^{t+\Delta t}, u_i^{t+\Delta t}$ are the displacement, velocity, and acceleration matrices, respectively. $\dot{u}_i^{t+\Delta t}, u_i^{t+\Delta t}$ can be calculated by the central difference method:

$$
\begin{aligned}
u_i^{t+\Delta t} &= u_i^t + \Delta t \dot{u}_i^t + \tfrac{1}{2} \ddot{u}_i^t \Delta t^2 \\
\dot{u}_i^{t+\Delta t} &= \dot{u}_i^t + \tfrac{1}{2}\Delta t \left( \ddot{u}_i^{t+\Delta t} + \ddot{u}_i^{t+\Delta t} \right)
\end{aligned} \tag{3}
$$

The position of the node can be expressed as

$$x^{n+1} = x^0 + u^{n+1} \tag{4}$$

The time step is calculated as follows:

$$\Delta t = \frac{L_{\min}}{c} \tag{5}$$

where $\Delta t$ is the time step, $L_{\min}$ is minimum element length, and $c$ is the sound speed.

A rigid body is an element of infinite stiffness defined on a number of nodes. Its most general movement consists of spatial rotations and translations. The links with the rest of the model are fixed. The motion of a rigid body is completely defined by the translations and rotations of its center of gravity (COG). The motion of the COG is monitored according to its own equations of motion. The motions of the individual nodes of the RB are then back-calculated.

### 2.2. Interactions between Elements

The non-penetration conditions are maintained by preventing each slave node in contact from crossing the respective master segment. The non-penetration condition of the contact boundary constraint is precisely defined in Equation (6).

$$g_N^t = g\left( ^{S}x^t, t \right) = \left( ^{S}x^t - ^{M}x^t \right) {}^{M}n^t \geq 0 \tag{6}$$

where $^{S}x^t$ and $^{M}x^t$ are the position at time $t$ of the node and any point on the main contact surface, and $^{M}n^t$ is the unit outward vector at the projection point. In each time step iteration, assuming that $^{M}n^t$ is a constant. The variation in the gap function $g_N^t$ is calculated to obtain a new displacement value, and then a new $^{M}n^t$ is calculated to replace the original value and the next iteration calculation is carried out.

The node equivalent force of the master flat shell elements can be obtained according to the principle of virtual work.

$$f_i = N^T F_i \tag{7}$$

where $f_i$ is the equivalent node force of the element, $F_i$ is the force of node on the contact element, and $N^T$ is the shape function of the element.

The force $F_i$ can be resolved into the normal component $f_s$ and the tangential component $f_c$, which are determined by the following equations.

$$F_i = f_s + f_c \tag{8}$$

$$f_s = |f_{se} + f_{cv}|n \tag{9}$$

$$f_c = min(|uf_s|, |f_{ce}|)t \tag{10}$$

where $f_{se}$ and $f_{cv}$ are the normal elastic and damping forces, respectively, $f_{ce}$ is the tangential elastic force, $u$ is friction coefficient between elements, and $n$ and $t$ are the tangential unit vector.

The nonlinear normal elastic force $f_{se}$ and normal viscous force $f_{sv}$ between elements are given by Equations (11) and (12), respectively,

$$f_{se} = k_{Ni}\delta_i = \left(1 + \frac{(\varepsilon - 1)\delta_i^2}{H_{cont}^2}\right)k_i\delta_i \tag{11}$$

$$f_{sv} = -2\xi_i v_i \sqrt{k_{Ni}m_i} \tag{12}$$

$$k_i = nk_{stable} \tag{13}$$

$$k_{stable} = \frac{m_1 m_2}{m_1 + m_2} \cdot \frac{1}{\Delta t^2} \tag{14}$$

$$\Delta t = L_c \sqrt{\frac{\rho}{E}} \tag{15}$$

where $k_i$ is the local contact stiffness, $\varepsilon$ is the parameter for nonlinear penalty stiffness, $\xi_i$ is the damping coefficient in the tangential direction, $m_i$ is the mass of nodes, and $v_i$ is the tangential resultant velocity on the contact point. $n$ is the scale factor for sliding interface penalties.

The tangential elastic force $f_{c,e}$ and tangential friction force $f_c$ between elements are as follows.

$$f_{ce} = \frac{3}{2(2-\nu)(1+\nu)}k_{Ni}\delta_j \tag{16}$$

$$f_c = uf_s \tag{17}$$

where $k_{Ni}$ is the normal stiffness, $\nu$ is Poisson's ratio, and $\delta_j$ is the tangential relative displacement at contact nodes, $u$ is the friction coefficient between elements.

## 3. Dynamic Model of Aircraft and Catapult System

To improve the dynamic analysis efficiency and achieve more accurate simulations of the complete catapult takeoff process in actual use, the following assumptions are adopted:

1.  The breaking pin of the holdback bar, the tire, and buffer of the landing gear are modeled as flexible bodies, and the other parts are rigid bodies. The multiple rigid bodies are connected through joints;
2.  The constraints between the internal members of the holdback bar and launch bar are regarded as ideal constraints;
3.  The piston, cylinder, and other hydraulic structures are modeled using rigid bodies;
4.  The deck turbulence that is perpendicular to the deck runway is neglected.

### 3.1. Configuration of Aircraft

The dynamic model presented in this paper encompasses the launch bar, holdback bar, shuttle, nose landing gear (NLG), main landing gear (MLG), fuselage, and carrier deck, as depicted in Figure 1. The deformation and stress of the UAV fuselage structure are not the main concern. Therefore, the aircraft fuselage is modeled using a rigid body. The influence of the engine rotational torque is neglected, and the engine thrust is decoupled

into three-axis forces acting at a point inside the aircraft fuselage. The relative positions of the aerodynamic forces and thrust with respect to the aircraft body are depicted in Figure 2.

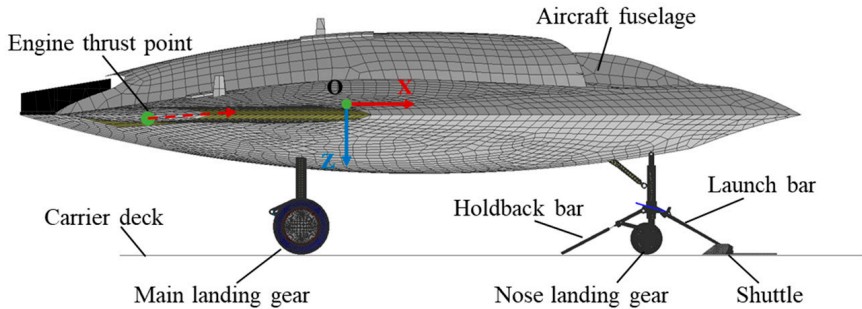

**Figure 1.** Schematic diagram of the aircraft.

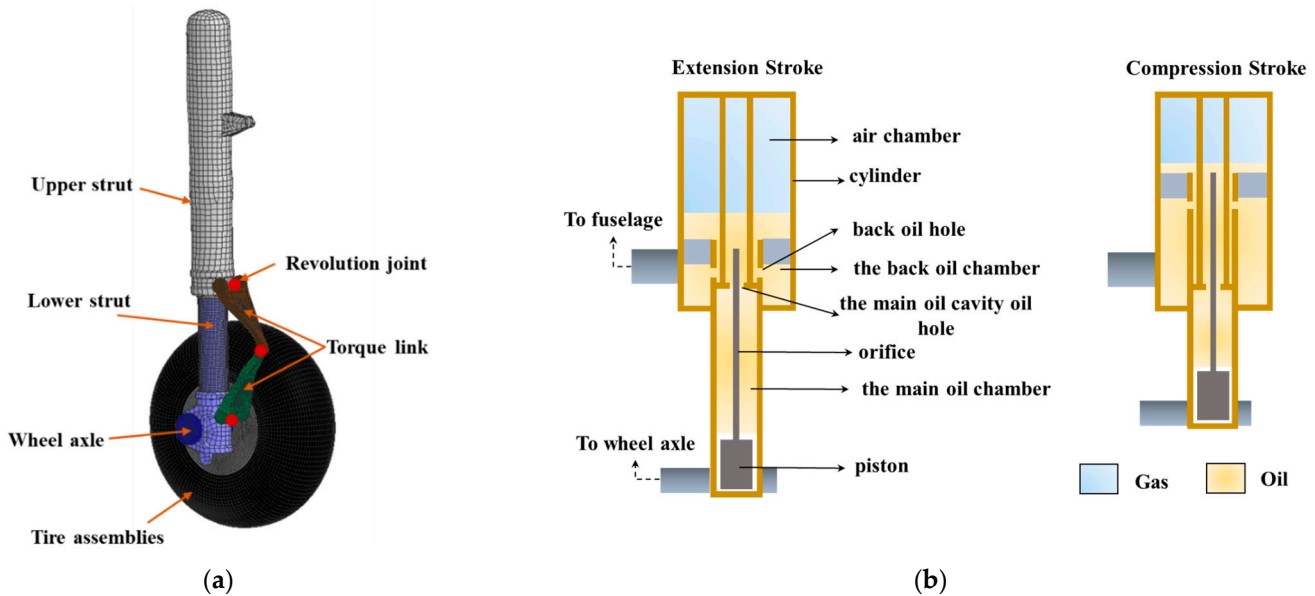

**Figure 2.** Dynamic model of main landing gear system: (**a**) diagram of MLG model; (**b**) diagram of damper in extension and compression stroke.

Aerodynamic forces are applied as six-degree-of-freedom loads (X, Y, Z, L, M, and N) on the fuselage rigid body, with the point of application being converted into the center of gravity, as shown in Figure 1. The fuselage rigid body includes the aircraft fuselage and components of the landing gear and establishes a constraint relationship with MLG and NLG in the form of contact. The values of X, Y, Z, L, M, and N are calculated using Equation (18).

$$
\begin{aligned}
\text{X} &= \tfrac{1}{2}\rho u^2 S_{ref} C_x + F_T \cos \alpha_T \\
\text{Y} &= \tfrac{1}{2}\rho u^2 S_{ref} C_y + F_T \sin \alpha_T \\
\text{Z} &= \tfrac{1}{2}\rho u^2 S_{ref} C_z \\
\text{L} &= \tfrac{1}{2}\rho u^2 S_{ref} b C_l \\
\text{M} &= \tfrac{1}{2}\rho u^2 S_{ref} c C_m + M_T \\
\text{N} &= \tfrac{1}{2}\rho u^2 S_{ref} b C_n
\end{aligned}
\tag{18}
$$

where $\rho$ is the density of air, $u$ is the aircraft speed, $F_T$ is the engine thrust, $M_T$ is the pitch moment produced by engine thrust, $\alpha_T$ is the tilt angle of the engine thrust line with respect to the fuselage coordinate system, and $C_x$, $C_y$, $C_z$, $C_l$, $C_m$, and $C_n$ are aerodynamic coefficients, which are the function of angle of attack, respectively. $c$, $b$, and $S_{ref}$ are the

reference chord, reference span, and reference wing area of the UAV, respectively. And the aircraft speed and angle of attack is calculated by the velocity vector of the fuselage rigid body at the center of gravity.

### 3.2. Dynamic Model of Main Landing Gear

The landing gear serves as the ground support system for the aircraft and functions as a pivotal energy-absorbing component during the landing process. As shown in Figure 2, the dynamic model of the landing gear consists of the upper strut, lower strut, torque link, wheel axle, and tire assemblies.

The displacement of the damper is determined by relative motion of the upper and lower struts. The hydraulic force $F_s$ can be expressed as

$$F_s = F_a + F_h \tag{19}$$

where $F_a$ is the air spring force and $F_h$ is the hydraulic damping force.

The air spring force [17] $F_a$ can be represented as

$$F_a = \begin{cases} A_a^L \left[ \dfrac{P_{a0}^L}{\left(1 - \frac{A_a^L S}{V_{a0}^L}\right)^{\gamma}} - P_{atm} \right], 0 < S \le S_{H0} \\ A_a^L \left[ \dfrac{P_{a0}^L}{\left(1 - \frac{A_a^L S}{V_{a0}^L}\right)^{\gamma}} - P_{atm} \right] + A_a^H \left[ \dfrac{P_{a0}^H - P_{a0}^L}{\left(1 - \frac{A_a^H (S - S_{H0})}{V_{a0}^H}\right)^{\gamma}} - P_{atm} \right], S > S_{H0} \end{cases} \tag{20}$$

where $A_a^L$ and $A_a^H$ are the piston area of the low- and high-pressure chamber, $P_{a0}^L$ and $P_{a0}^H$ are the initial pressure of the low- and high-pressure chamber, $V_{a0}^L$ and $V_{a0}^H$ are the initial buffer filling volume of the low- and high-pressure chamber, $P_{atm}$ is the atmospheric pressure, $S$ is the buffer stroke, $S_{H0}$ is the initial stroke of the high-pressure chamber, and $\gamma$ is the variable gas index.

The air spring force varies between the low-pressure and high-pressure chambers. The hydraulic damping force can be expressed as follows,

$$F_h = \begin{cases} \dfrac{\rho_h A_h^3}{2\left(C_d^+\right)^2 A_d^2} \dot{S}^2 + \dfrac{\rho_h A_{hl}^3}{2\left(C_{dl}^+\right)^2 \left(A_{dl}^+\right)^2} \dot{S}^2, \dot{S} \ge 0 \\ -\dfrac{\rho_h A_h^3}{2\left(C_d^-\right)^2 A_d^2} \dot{S}^2 - \dfrac{\rho_h A_{hl}^3}{2\left(C_{dl}^-\right)^2 \left(A_{dl}^-\right)^2} \dot{S}^2, \dot{S} \le 0 \end{cases} \tag{21}$$

where $\rho_h$ is the oil density, $\dot{S}$ is the stroke velocity, $A_h$ is the effective area of the buffer, $A_d$ is the main oil cavity oil hole area, $C_d^+$ and $C_d^-$ are the flow coefficient of the main oil hole at the forward and reverse stroke, $A_{hl}$ is the effective area of the back oil hole, $A_{dl}^+$ and $A_{dl}^-$ are the effective flow area of the oil return hole at the compression and reverse stroke, and $C_{dl}^+$ and $C_{dl}^-$ are the flow coefficient of the back oil hole at the compression and reverse stroke.

Besides the shock absorber load, the flexibility of the tire also contributes significantly to the impact load during the aircraft catapult. The compression of the tire under impact load is a significant proportion of the overall compression stroke of the landing gear damping system. The internal structure of the tire is depicted in Figure 3a, where the inner layer of the tire is defined as the fabric material and the volume surrounded by the wheel rim and the inner fabric layer of the tire is filled with gas.

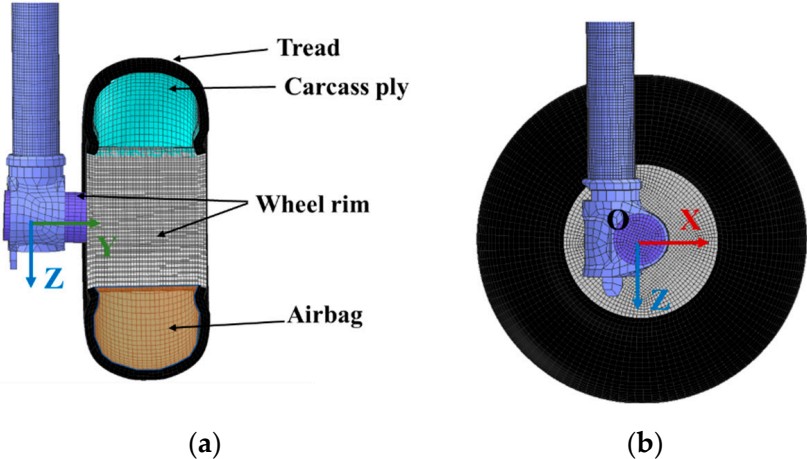

(a)　　　　　　　　　　　(b)

**Figure 3.** Dynamic model of flexible tire: (**a**) Internal structure of tire; (**b**) Constraint of tire and axle.

The tread and the wheel rim share common nodes on the adjacent surface, and the rotational constraints of the tire are defined using a coordinate system O-XYZ located at the center of the wheel rim, as shown in Figure 3b. The deformation of the tire can be approximated as an adiabatic process with an adiabatic parameter set at 1.4. The pressure inside the tire at time $t_{n+1}$ can be expressed as

$$P_{n+1} = P_n \left( \frac{V_n}{V_{n+1}} \right)^{1.4} \tag{22}$$

where $V_n$ and $P_n$ are the volume and pressure at time step $n$, respectively, and $V_{n+1}$ and $P_{n+1}$ are the volume and pressure at time step $n+1$.

The rubber material of the tire is represented using eight-node hexahedral elements and is characterized by the Mooney–Rivlin material model [18]. The constitutive equation for this model is given by

$$W = A(I - 3) + B(II - 3) + C\left(III^{-2} - 1\right) + D(III - 1)^2 \tag{23}$$

where $C = 0.5A + B$, $D = \frac{A(5v-2)+B(11v-5)}{2(1-2v)}$, A, and B are constants determined through the experiment. $v$ is the Poisson's ratio, and I, II, and III are Green–Lagrange strain tensor constants.

### 3.3. Catapult Launch System

In this study, we establish a model for the electromagnetic catapult. In this paper, the Electromagnetic Aircraft Launch System (EMALS) model is established. Figure 4 shows the carrier-based aircraft's nose landing gear in the tensioning process. The rear side of the nose landing gear is connected to the deck through the holdback fitting, release element, and holdback bar. The ejection force is exerted on the shuttle, and it is transmitted to the aircraft's nose landing gear through the catapult bar [17].

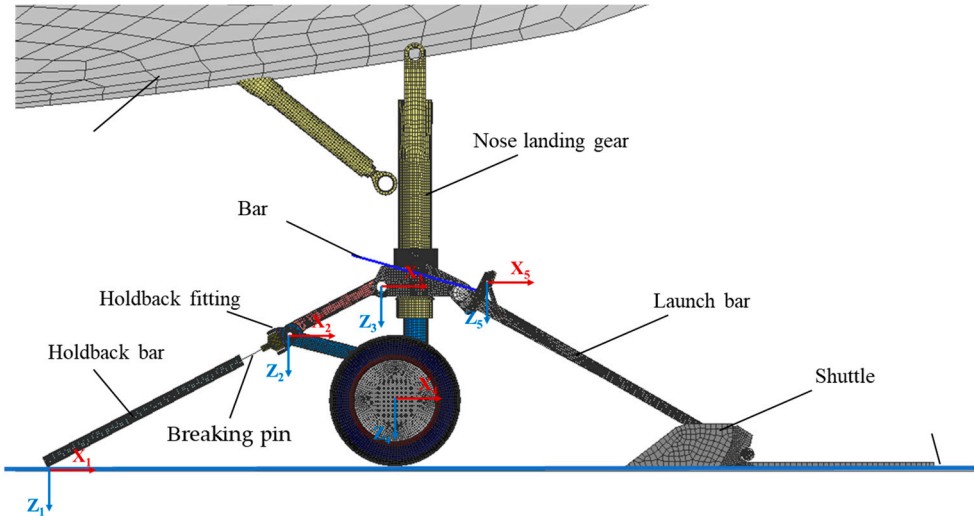

**Figure 4.** Engagement system of NLG, launch bar and holdback bar.

### 3.3.1. Ejection Force

The model of the linear motor was developed based on the structural characteristics of the permanent magnet linear synchronous motor [19]. This model neglects the saturation of the iron core, as well as the eddy current and hysteresis losses. The ejection force is given by

$$
\begin{aligned}
F &= \frac{m_e E_l}{v_e \left( R_l^2 + X^2 \right)} \left( XU \sin\theta + R_l \cos\theta - E_l R_l \right) \\
\sin\theta &= \frac{X}{\sqrt{R_l^2 + X^2}} \\
\cos\theta &= \frac{R_l}{\sqrt{R_l^2 + X^2}}
\end{aligned}
\tag{24}
$$

where $X$ is the impedance of armature winding of equivalent circuit, $U$ is the voltage applied to the armature winding, $R_l$ is the per phase resistance of armature winding, $m_e$ is the mass of shuttle, $E_l$ is the excitation potential, and $v_e$ is the initial speed of the linear motor. The excitation potential and the initial speed of the linear motor are calculated by

$$
\begin{aligned}
E_l &= 2\sqrt{2} f (N_w K_w) \tau b_e \overline{B} \\
v &= 2\tau f
\end{aligned}
\tag{25}
$$

where $N_w$ is the number of turns per phase, $K_w$ is the effective turn coefficient for each phase of armature winding, $\tau$ is the pole distance, $b_e$ is the width of the permanent magnet, $\overline{B}$ is the average magnetic flux produced by the permanent magnetic field in the height range of the groove winding, and $f$ is the frequency of the motor.

### 3.3.2. Holdback System

The holdback bar lock mechanism needs to be constructed using materials with a higher yield strength. In this paper, an enhanced plasticity algorithm that includes transverse shear effects is used. The transverse shear effects exactly satisfy Hill's criterion [20] and precisely update the element thickness during plastic deformation. The material has isotropic elastic properties, as defined in Table 1.

**Table 1.** Material properties [21].

| | Mass Density (kg/m³) | Young's Modulus (MPa) | Yield Stress (MPa) | Poisson's Ratio |
|---|---|---|---|---|
| Alloy steel | 2700 | 70,000 | 450 | 0.3 |

In addition, an isotropic damage law is added to the basic elastic–plastic formulation. In that case, the equivalent stress σ is defined in the function of the equivalent total strain $\varepsilon_{eq}$ at each point in the element thickness including transverse shear effects:

$$\varepsilon_{eq} = \frac{1}{1+\mu}\left[\frac{1}{2}(\varepsilon_{11}-\varepsilon_{22})^2 + \frac{1}{2}(\varepsilon_{11}-\varepsilon_{33})^2 + \frac{1}{2}(\varepsilon_{22}-\varepsilon_{33})^2 + 3\varepsilon_{12}{}^2 + 3\varepsilon_{13}{}^2 + 3\varepsilon_{23}{}^2\right]^{1/2} \quad (26)$$

where $\varepsilon_{11}$, $\varepsilon_{22}$ and $\varepsilon_{33}$ are the normal strain of the element, and $\varepsilon_{12}$, $\varepsilon_{13}$, $\varepsilon_{23}$ are the shear strain of the element. $\mu$ is the Poisson's ratio. If the equivalent strain of the element maximum over thickness reaches one of the specified criteria, the element resistance is removed from the simulation, while its mass is conserved. And the element elimination is performed gradually over a time interval of 100 time steps.

## 4. Simulation and Results

Based on the rigid–flexible coupling dynamic model established previously, the dynamic responses of the catapult process at different launch bar angles and holdback bar release thresholds are analyzed. In this paper, the holdback bar release threshold is adjusted by the fracture strain of the holdback bar $\varepsilon_m$. The simulation time of the catapult take off process is 3.8 s. The values of different simulation conditions are shown in Table 2. The control of variables is conducive to the comparative study of the similarities and differences between multiple sets of aircraft attitudes.

**Table 2.** Initial simulation conditions of aircraft in each case.

| Value | Case | | | | | | | |
|---|---|---|---|---|---|---|---|---|
| | 1 | 2 | 3 | 4 | 5 | 6 | 7 | 8 |
| Angle of launch bar (°) | 35 | 40 | 45 | 50 | 40 | 40 | 45 | 45 |
| Fracture strain of the holdback bar $\varepsilon_m$ | 0.3 | 0.3 | 0.3 | 0.3 | 0.28 | 0.32 | 0.28 | 0.32 |

### 4.1. Influence of the Launch Bar Angle on the Catapult Process

In this section, the influence of the launch bar angle on the catapult process is studied. In the numerical simulation calculation, the initial condition of the aircraft is represented in Table 2, and the launch bar angle $\varphi$ are, respectively, set at 35°, 40°, 45°, and 50°. The entire catapult process can be primarily divided into three phases: the initial 0.3 s involves a gradual tensioning process of the holdback bar, followed by the aircraft taxiing from 0.3 s to 3.4 s, and finally the nose landing gear extends at 3.4 s to enable the aircraft to increase its pitch angle and lift, thus taking off from the carrier deck. The holdback bar connects the nose landing gear to the deck. As the catapult load gradually increases, reaching the maximum limit of the holdback bar, the release element is eliminated. And the aircraft is no longer constrained by the holdback bar. Throughout the taxiing process, the center of mass position remains nearly constant.

As shown in Figure 5a, the launch bar angle has little influence on the acceleration of the aircraft's center of gravity during the catapult process. When the launch bar angle is 40°, the acceleration during the 2–3 s period is also greater than the other conditions, and there is the greatest fluctuation in acceleration during the 2.5 to 3-s phase. Figure 5b depicts the speed in the height of the aircraft's center of gravity during the catapult process. In the first 0.3 s, under the action of the holdback bar, the load acting on the nose landing gear pushes the entire carrier-based aircraft downward. After 0.3 s, the carrier-based aircraft, under the effect of the ejection force, begins to accelerate along the deck, and the fluctuation in vertical velocity gradually increases. Additionally, with a larger launch bar angle, the climb speed at the end of the catapult is also higher. However, when the launch bar angle reaches 50°, the climb speed at the end of the catapult process decreases.

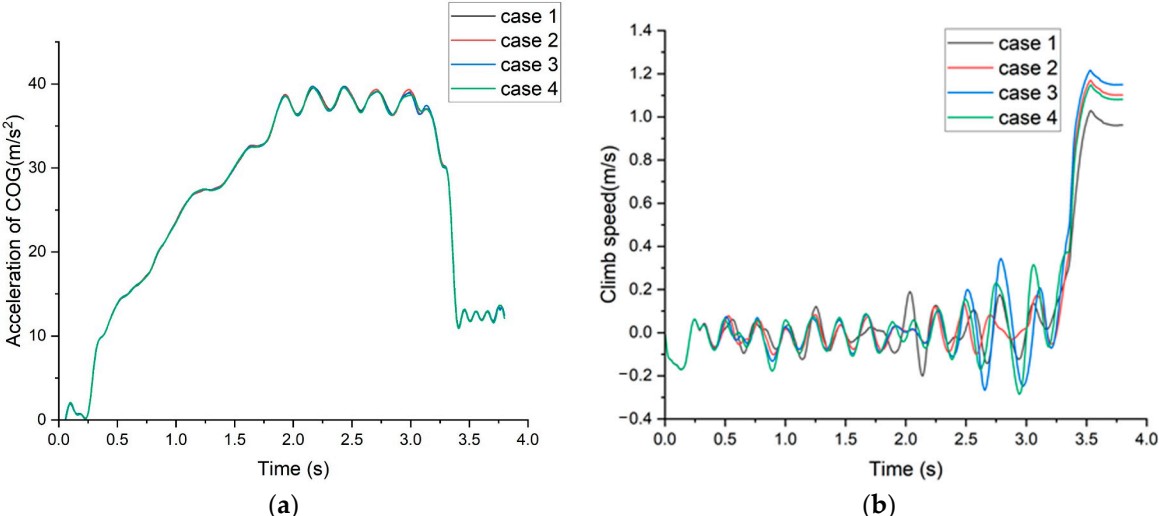

**Figure 5.** The acceleration and climb speed of carrier-based aircraft: (**a**) acceleration; (**b**) climb speed.

The contact force of the landing gear is depicted in Figure 6. In the tensioned state, the launch bar restrains the carrier-based aircraft on the carrier deck. Under the combined action of holdback bar and launch bar, the nose landing gear experiences higher loads, while the main landing gear experiences a smaller increase in contact load compared to the nose landing gear. When the holdback bar is released, the potential energy stored during the holdback bar tensioning phase is released. During the taxiing process, the height of the center of mass remains nearly constant. The launch bar angle significantly affects the load on the nose landing gear. A greater launch bar angle results in increased contact load on the nose landing gear and more significant load fluctuations.

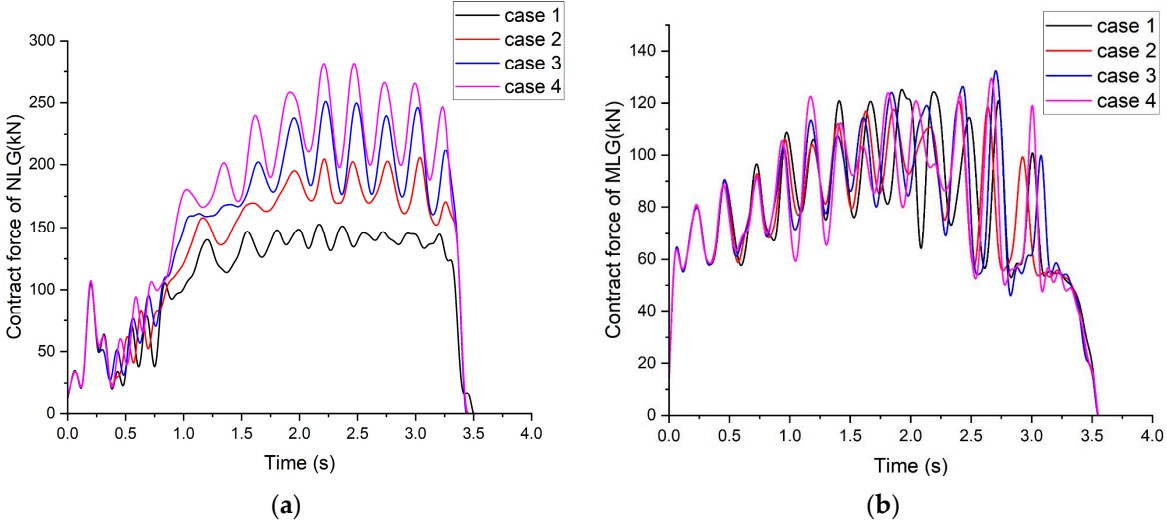

**Figure 6.** Time history of the contact force of landing gear: (**a**) NLG; (**b**) MLG.

Figure 7 depicts the change in the elongation of the aircraft's landing gear damper during the catapult process. In the initial 0.3 s, during the tensioned state, the holdback bar restrains the carrier-based aircraft on the carrier deck. This causes the nose landing gear damper to compress, and simultaneously, the compression of the main landing gear increases. Under the same ejection force, a larger launch bar angle results in a greater compression of the nose landing gear during the catapult process. Therefore, between 3.4 and 3.7 s, the rebound of the nose landing gear is also greater.

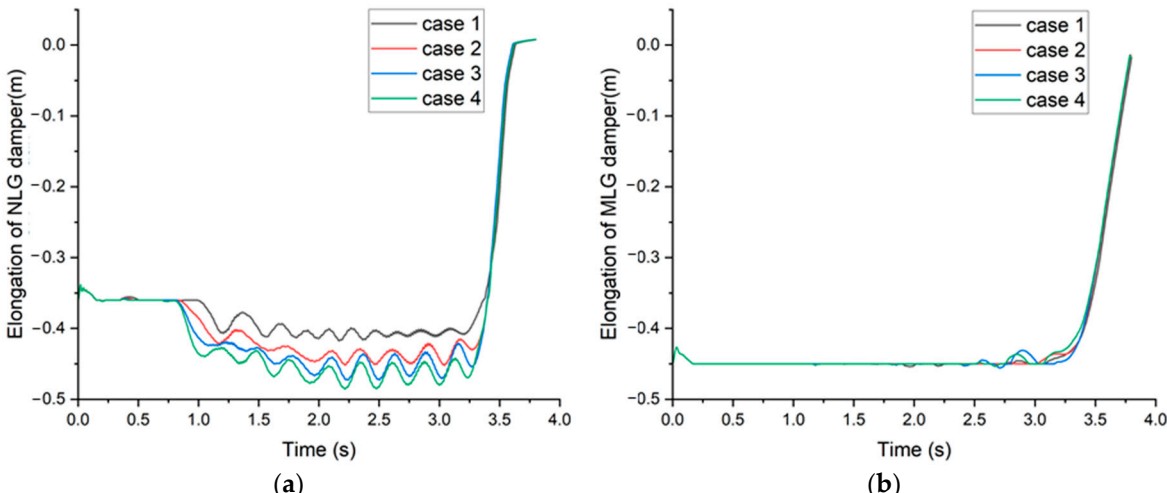

**Figure 7.** Time history of the elongation of landing gear damper: (**a**) NLG; (**b**) MLG.

As shown in Figure 8a, the interference caused by the holdback bar angle on the aircraft pitch angle is significant. During the catapult process within 1 s, under the influence of the launch bar, the carrier-based aircraft experiences a decrease in pitch angle. Moreover, with a larger launch bar angle, the reduction in pitch angle is more pronounced. Between 1 s and 3.4 s, the pitch angle fluctuation decreases. Additionally, a larger angle of the catapult bar results in greater fluctuations in pitch angle during this period. After 3.4 s into the catapult process, the aircraft's pitch angle rapidly increases. When the launch bar angle is set to 45°, the aircraft reaches its maximum pitch angle at 3.8 s into the catapult process. Figure 8b illustrates the variation in pitch angle rate during the catapult process.

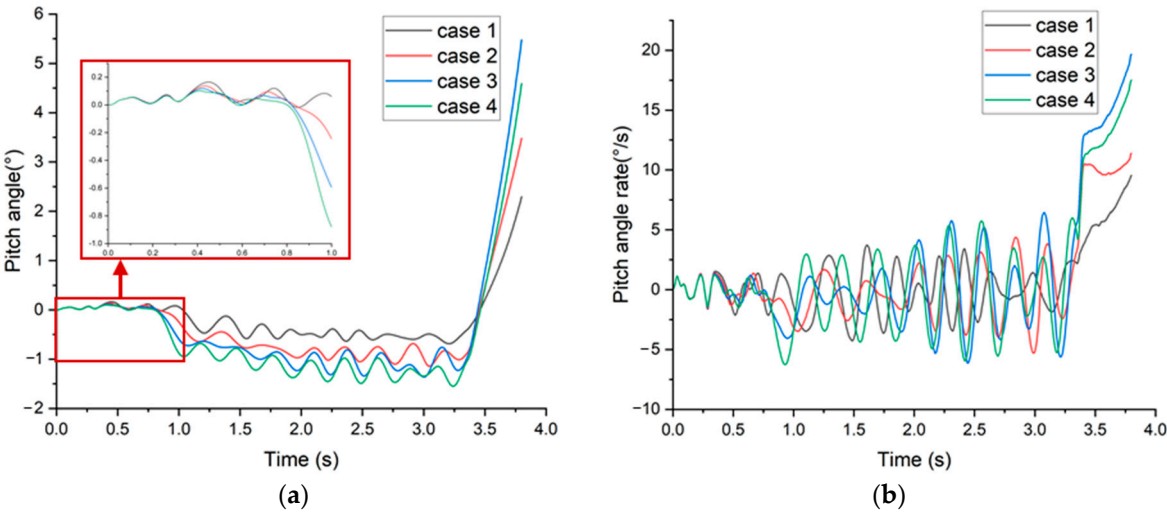

**Figure 8.** Pitch angle and pitch angle rate during the catapult process: (**a**) pitch angle; (**b**) pitch angle rate.

The trend in angle rate changes more effectively reflects the influence of the launch bar angle on the aircraft's pitch attitude during the catapult process. Specifically, when the launch bar angle is larger, there is a greater variation in pitch angle rate during the taxi phases. The launch bar angle has a significant impact on the pitch angle rate during the extension phase of the nose landing gear. As the launch bar angle increases, the pitch angle rate also increases. However, when the launch bar angle is set to 50°, the pitch angle rate during the extension phase of the nose landing gear is lower than that when the launch bar angle is 45°.

## 4.2. Influence of Holdback Bar Release Threshold on the Catapult Process

The launch bar angles of aircraft are 40° and 45°, and the fracture strains are, respectively, set as 0.27, 0.3, and 0.33. As shown in Figure 9a,b, it can be observed that the fracture strain primarily affects the acceleration changes in the first 0.6 s of the catapult process, particularly during the catapult taxiing phase. With increased fracture strain, the initial aircraft acceleration upon fracture is higher, and this effect is more pronounced in cases with a launch bar angle of 45° compared to 40°.

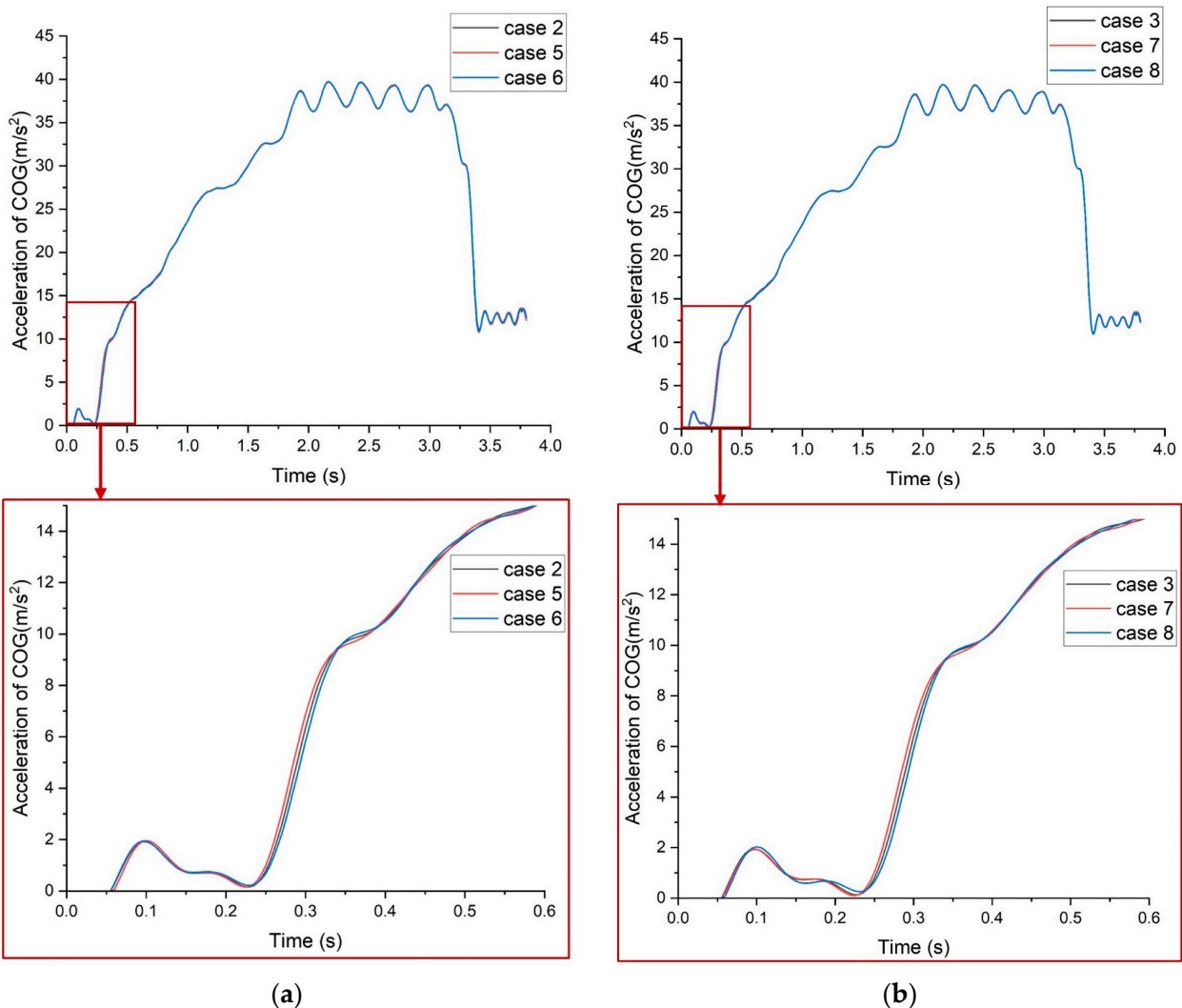

(**a**)                                        (**b**)

**Figure 9.** Time history of the aircraft acceleration with different holdback bar load thresholds: (**a**) launch bar angle = 40°; (**b**) launch bar angle = 45°.

The aircraft climb speed under different holdback bar fracture strains for launch bar angles of 40° and 45° is illustrated in Figure 10. During the holdback tensioning process, the aircraft's center of gravity shifts downward. In the subsequent catapult taxi phase, the aircraft's center of gravity fluctuates up and down around the initial position, with larger movements occurring as the taxi time increases. The holdback bar fracture strain has little impact on the aircraft's climb speed at the end of the catapult process.

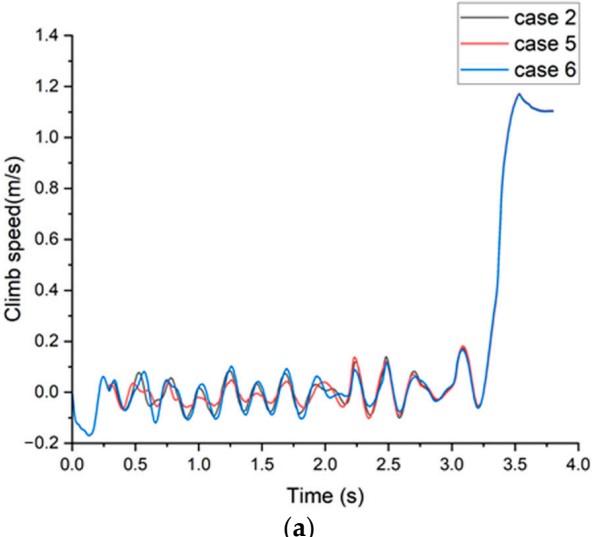
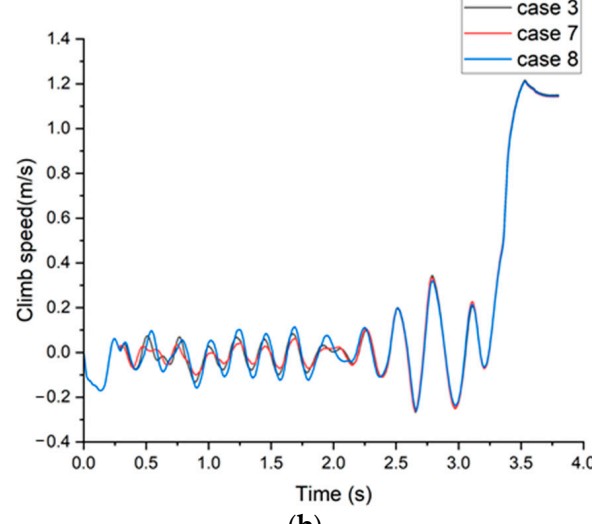

**Figure 10.** Time history of the aircraft climb speed with different holdback bar load thresholds: (**a**) launch bar angle = 40°; (**b**) launch bar angle = 45°.

The time history of the contact force of the nose landing gear and main landing gear is shown in Figure 11. It can be observed that the fracture strain of the holdback bar has a significant impact on the contact force of the nose landing gear from 0.3 to 1 s. An increased fracture strain results in higher contact forces on the nose landing gear upon release of the holdback. However, during the first 2 s of the catapult process, the NLG contact force of the condition with a fracture strain of 0.27 will be higher than the two conditions with fracture strains of 0.3 and 0.33. Figure 11b,d shows that the fracture strain of the holdback bar has an impact on the MLG contact force of the aircraft from 0.3 to 1 s of the catapult process. In this phase, a lower strain results in a lower MLG contact force and decreased load fluctuations.

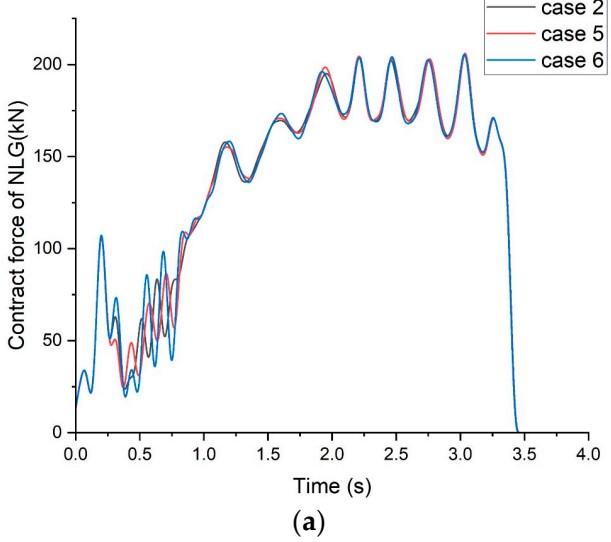
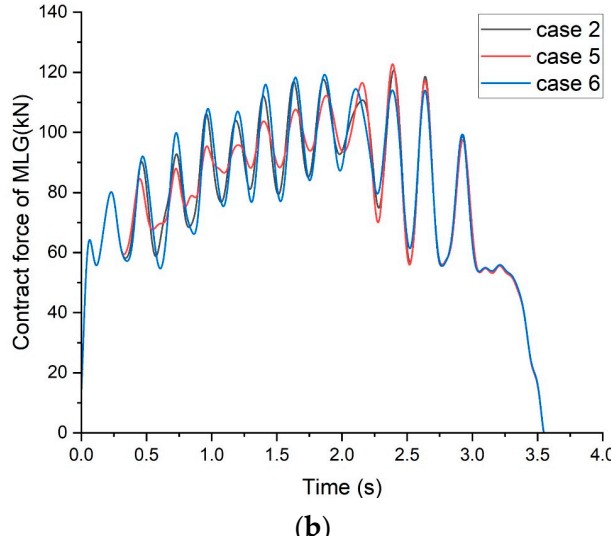

**Figure 11.** *Cont*.

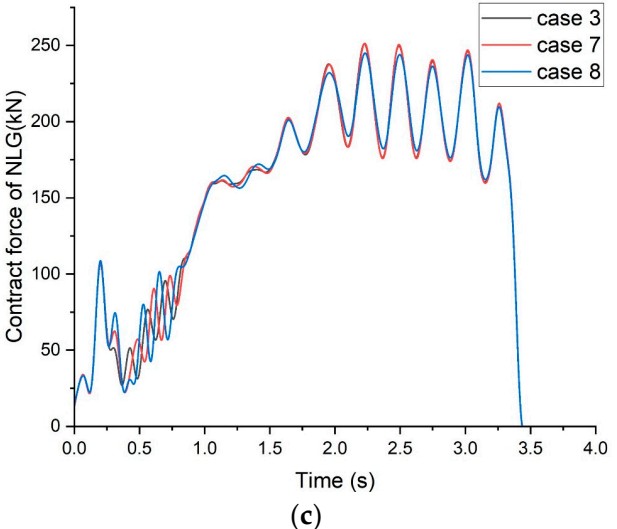

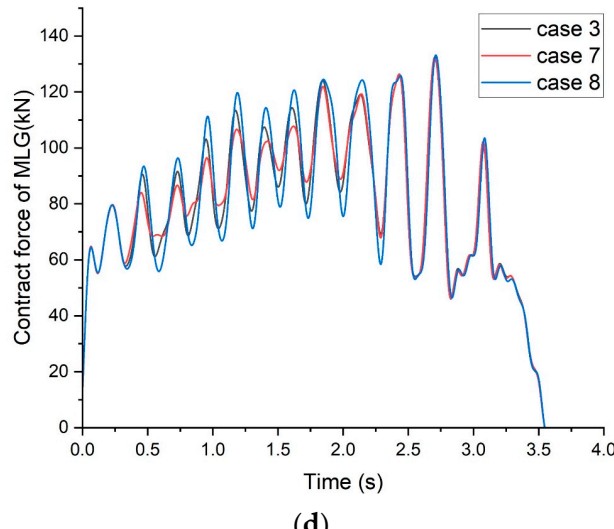

(**c**)

(**d**)

**Figure 11.** Time history of the contact force of landing gear with different holdback bar load thresholds: (**a**) pitch angle at launch bar angle = 40°; (**b**) pitch angle rate at launch bar angle = 40°; (**c**) pitch angle at launch bar angle = 45°; (**d**) pitch angle rate at launch bar angle = 45°.

Figure 12 depicts the variation in the landing gear damper elongation with different holdback bar load thresholds. As shown in Figure 12a,c, the strain of the holdback bar has a significant impact on the elongation of the NLG damper from 0.3 to 0.6 s into the catapult process. After the holdback bar releases, the NLG damper rebounds, and the greater the fracture strain, the greater the rebound of the NLG. It can be seen that the fracture strain has little effect on the elongation of the MLG, as shown in Figure 12b,d.

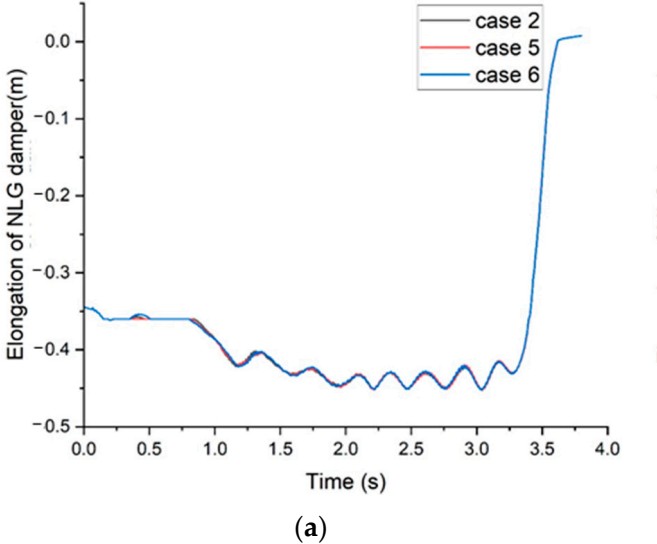

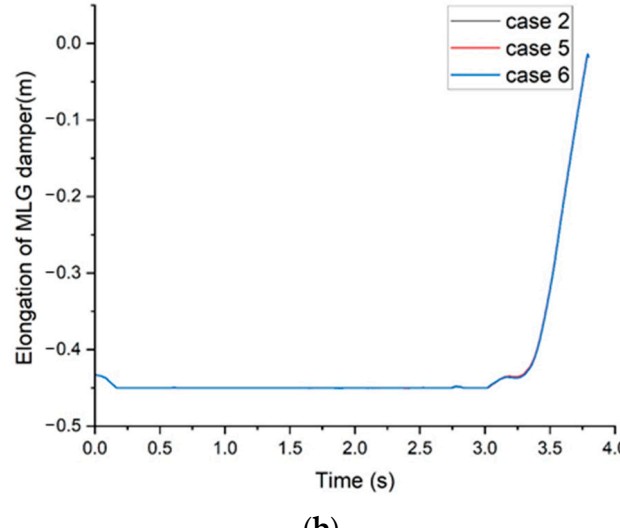

(**a**)

(**b**)

**Figure 12.** *Cont.*

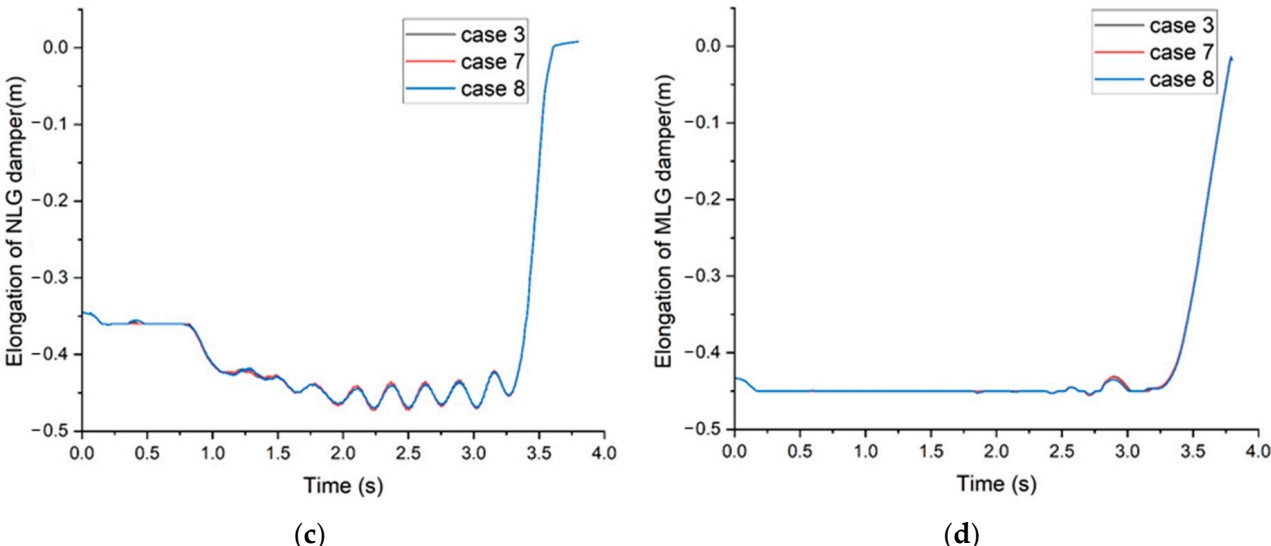

**Figure 12.** Time history of the elongation of landing gear damper: (**a**) $\varepsilon_m = 40°$, NLG; (**b**) $\varepsilon_m = 40°$, MLG; (**c**) $\varepsilon_m = 45°$, NLG; (**d**) $\varepsilon_m = 45°$, MLG.

Different fracture strains of the holdback bar have an impact on the pitch attitude of the carrier-based aircraft during the catapult process, as shown in Figure 13. It can be seen that the fracture strain of the holdback bar has a significant impact on the pitch angle of the carrier-based aircraft in the first second of the catapult process. Higher fracture strains lead to more significant pitch angle variations. During the taxiing process, the carrier-based aircraft maintains a negative pitch angle. At the end of the catapult process, when the fracture strain is 0.27, the aircraft achieves a larger pitch angle than the other two conditions. The variation in pitch angle rate confirms this pattern, as shown in Figure 13b,d. Different fracture strains of the holdback bar have the greatest impact on the pitch angle rate of the carrier-based aircraft during the first 0.3 to 1 s of the catapult-assisted takeoff. Larger fracture strains result in higher pitch angle velocities during the catapult process. Between 2.5 and 3.8 s of the catapult process, lower fracture strains result in higher pitch angle velocities.

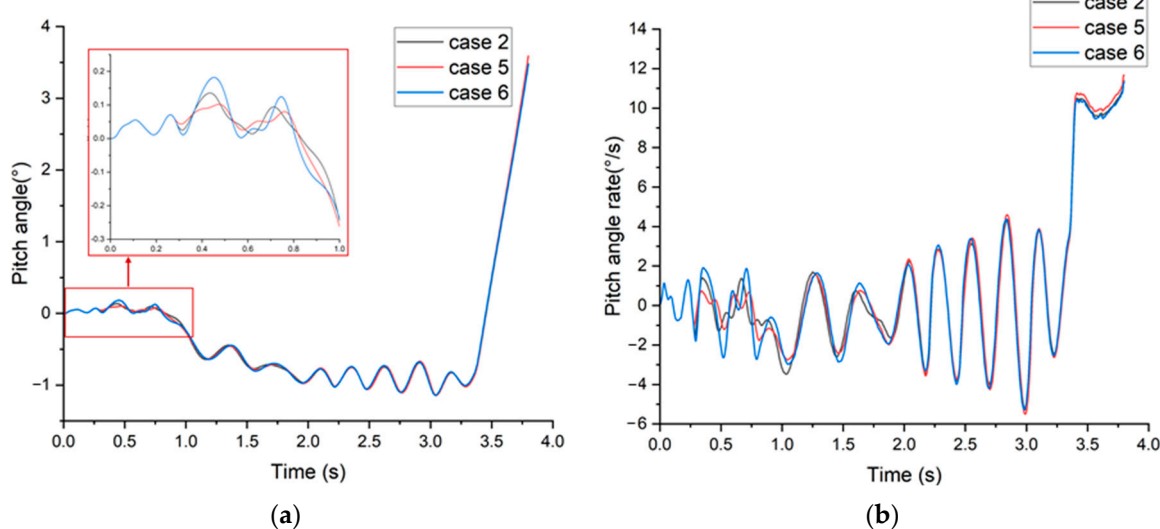

**Figure 13.** *Cont.*

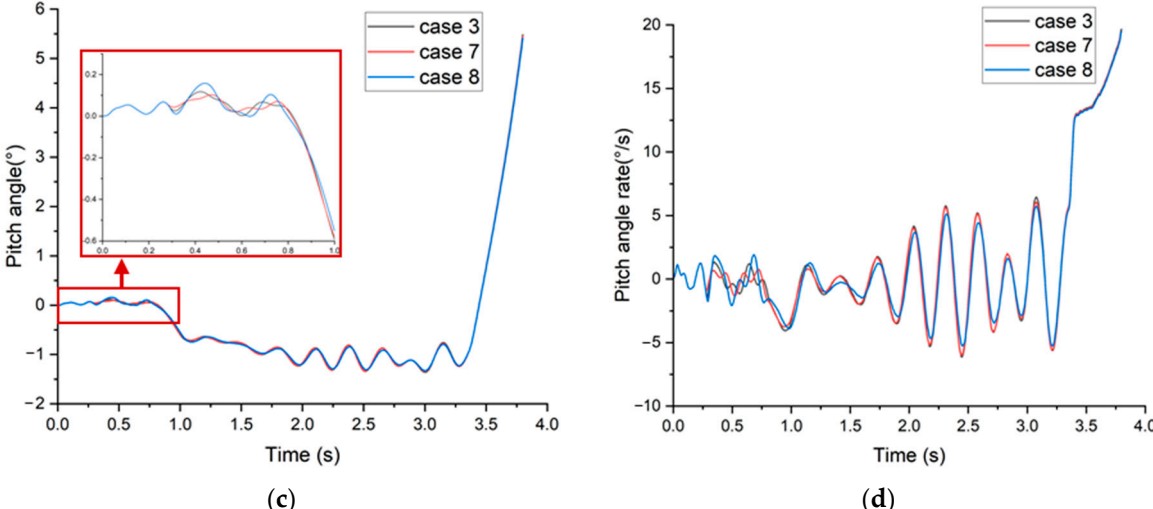

**Figure 13.** Pitch angle and angle rate during the catapult process: (**a**) pitch angle at launch bar angle = 40°; (**b**) pitch angle rate at launch bar angle = 40°; (**c**) pitch angle at launch bar angle = 45°; (**d**) pitch angle rate at launch bar angle = 45°.

## 5. Conclusions

Carrier-based aircraft catapult-assisted takeoff is a complex process, and any issues during the catapult process at any phase can potentially lead to catastrophic accidents. The analysis conducted in this paper takes into account various factors, including the material properties of the launch bar, the characteristics of the landing gear system, and the state of the aircraft. Simulations are performed to analyze the catapult process under different launch bar angles and holdback bar release thresholds.

1. This paper establishes a dynamic model for a certain carrier-based aircraft based on the FEM-MBD method. The simulation analysis of the catapult-assisted takeoff process aims to resolve the problems of the coupling among multi-motion bodies and flight environments. The catapult-assisted takeoff process consists of three phases: the tensioning and release of the holdback bar, carrier deck taxiing, and extension of the nose landing gear. In the simulation scenarios presented in this paper, the first 0.3 s represent the holdback bar tensioning process, from 0.3 s to 3.4 s is the carrier-based aircraft's deck taxiing, and from 3.4 s to 3.8 s, the nose landing gear extends and the carrier-based aircraft takes off.

2. In the cases of 35°, 40°, 45°, and 50° launch bar angles, simulation results under different launch bar angles indicate that the launch bar angle has a significant impact on the climb speed at the end of the catapult process, the load and extension of the nose landing gear, and the aircraft's pitch attitude. As the launch bar angle increases, the load on the nose landing gear increases, and the load fluctuations become more pronounced. Under the same catapult load, a larger launch bar angle results in greater compression of the nose landing gear during the catapult process. Consequently, the rebound of the nose landing gear between 3.4 to 3.7 s is also greater. During the catapult process from 1 s to 3.4 s, there are decreased fluctuations in the pitch angle, and the larger the launch bar angle, the greater the amplitude of pitch angle fluctuations. After 3.4 s into the catapult process, the aircraft's pitch angle rapidly increases. When the launch bar angle is 45°, the maximum pitch angle of the aircraft is reached at 3.8 s. The launch bar angle has a significant impact on the rate of the pitch angle during the extension of the nose landing gear. A larger launch bar angle results in a higher pitch angle rate. However, when the launch bar angle is 50°, the pitch angle rate during the extension of the nose landing gear is lower compared to the scenario with a 45° launch bar angle.

3. The comparison of simulation results for six different scenarios, involving three different holdback bar release thresholds each for launch bar angles of 40 degrees and 45 degrees, shows that the model can reasonably capture the dynamic characteristics of the

carrier-based aircraft catapult process. The holdback bar fracture strain has a significant impact on the pitch angle during the first second of the aircraft catapult process, with greater holdback bar fracture strain leading to larger pitch angle variations. A higher launch bar fracture strain results in a more pronounced change in the pitch attitude of the carrier-based aircraft during 0.3 to 1 s in the catapult process. This paper demonstrates that the method utilizing the central difference method for solving the coupled rigid–flexible finite element model can still be effectively used to simulate the catapult process. This method provides a better representation of the aircraft's attitude changes during the catapult process.

**Author Contributions:** Conceptualization, H.S. and Z.K.; methodology, H.S.; software, H.S. and D.L.; validation, H.S., Z.K. and C.W.; formal analysis, H.S. and Z.K.; investigation, H.S., J.X. and S.Z.; resources, Z.K. and C.W.; data curation, H.S. and Z.K.; writing—original draft preparation, H.S.; writing—review and editing, H.S. and S.Z.; visualization, H.S. and Z.K.; supervision, D.L.; project administration, D.L.; funding acquisition, J.X. and D.L. All authors have read and agreed to the published version of the manuscript.

**Funding:** This research was funded by the National Natural Science Foundation of China, no. T2288101, and the National Key Research and Development Project, grant number 2020YFC1512500.

**Data Availability Statement:** Data are contained within the article.

**Conflicts of Interest:** Author Chunsheng Wang was employed by the company China Aero Polytechnical Establishment. The remaining authors declare that the research was conducted in the absence of any commercial or financial relationships that could be construed as a potential conflict of interest.

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
