# Peer review of "Analysis of Catapult-Assisted Takeoff of Carrier-Based Aircraft Based on Finite Element Method and Multibody Dynamics Coupling Method"

_aerospace, doi:10.3390/aerospace10121005_

Round 1

Reviewer 1 Report

Comments and Suggestions for Authors

This paper developed a rigid-flexible coupling model using the finite element method and multibody dynamics approach, which was used to analyze the catapult-assisted takeoff of an UAV for several different cases.

 There are several comments need to address:

1.      The last sentence in the abstract should be rewritten, as the self-evaluation like “offer valuable insights into…” is not very proper here, instead, its better to use data to prove this.

2.      Where are the references 1-3 in Introduction?

3.      Whenever possible, the original source of the equations and data should be properly mentioned/cited. For example, where do the Eqs. 20, 23 and Table 1 come from?

4.      The title of the Table 2 is missing.

5.      You mentioned that the aerodynamic forces are applied as 6 DOFs… But how do you get the aerodynamic loads as this is also very critical for this simulation?

6.      During the takeoff acceleration process, the speed, AOA, etc. are changing, so does the aerodynamic force generated by the UAV, but how do the aerodynamic loads change during the process is missing.

Comments on the Quality of English Language

In general is fine, but can be further improved.

Author Response

Dear reviewer,

We sincerely thank you for the valuable feedback that we have used to improve the quality of our manuscript. The reviewer comments are laid out below in normal font and our response is given in red font.

Reviewer 2 Report

Comments and Suggestions for Authors

The authors present a nice paper on the numerical modelling of dynamic forces, accelerations and deflections on an aircraft landing gear structure when the aircraft is launched by catapult.

The background adequately covers the historical background and the numerical approach seems sound. However there is no validating experiment, leaving the numerical computation in a state of unknown accuracy. The model doend need to be validated against a real carrier catapult, but a lab setup should suffice.

the words "Tensioned" on row 385 has a differend font in the "...ed"

Author Response

Dear editors and reviewers,

We sincerely thank you for the valuable feedback that we have used to improve the quality of our manuscript. The reviewer comments and our response is given in the attachment file.

Round 2

Reviewer 1 Report

Comments and Suggestions for Authors

Thanks for the revision, which makes the manuscript looks better now. One more comment from my side is:

 - In the last part of your abstract, you emphasize the capabilities of your method, but fail to present the findings of this paper. As a reader, after reading your abstract, I did not sense any quantifiable improvement in your model.

Comments on the Quality of English Language

Fine

Author Response

    Thanks for your careful review and guidance in this article. We have added key conclusions to the abstract. The relevant content is highlighted in red.

Reviewer 2 Report

Comments and Suggestions for Authors

Review acknowledged

Author Response

Dear reviewer,

    Thank you for your time and feedback on my manuscript. We welcome your continued cooperation.